# The Bioaccessibility of Phenolics, Flavonoids, Carotenoids, and Capsaicinoid Compounds: A Comparative Study of Cooked Potato Cultivars Mixed with Roasted Pepper Varieties

**DOI:** 10.3390/foods10081849

**Published:** 2021-08-11

**Authors:** Mansor Hamed, David G. Holm, Michael Bartolo, Pinky Raigond, Vidyasagar Sathuvalli, Sastry S. Jayanty

**Affiliations:** 1San Luis Valley Research Center, Department of Horticulture and Landscape Architecture, Colorado State University, 0249 East County Road 9N Center, Fort Collins, CO 81125, USA; Mansor.Hamed@rams.colostate.edu (M.H.); David.Holm@colostate.edu (D.G.H.); 2Arkansas Valley Research Center, Department of Horticulture and Landscape Architecture, Colorado State University, 27901 County Rd 21, Rocky Ford, CO 81067, USA; Michael.Bartolo@ColoState.EDU; 3Crop Physiology, Biochemistry & Post-Harvest Technology Division, ICAR-Central Potato Research Institute, Shimla 171001, India; jariapink@gmail.com; 4Hermiston Agricultural Research & Extension Center, Oregon State University, 2121 S 1st Street, Hermiston, OR 97838, USA; Vidyasagar@oregonstate.edu

**Keywords:** pepper, potato, bioaccessibility, bioactive compounds, capsaicin

## Abstract

An in vitro method was used to assess the bioaccessibility of phenolics, flavonoids, carotenoids, and capsaicinoid compounds in different cooked potatoes mixed with roasted peppers (*Capsicum annuum*), Joe Parker (JP, hot), and Sweet Delilah (SD, sweet). The present study identified differences in the bioaccessibility of bioactive compounds among the potato cultivars (*Solanum tuberosum*) Purple Majesty (PM; purple flesh), Yukon Gold (YG; yellow flesh), Rio Grande Russet (RG; white flesh) and a numbered selection (CO 97226-2R/R (R/R; red flesh)). The bioactive compounds and capsaicinoid compounds in potatoes and peppers were estimated before and after in vitro digestion. Before digestion, the total phenolic content of potato cultivars mixed with JP was in the following order: R/R > PM > YG > RG. The highest levels of carotenoids were 194.34 µg/g in YG and 42.92 µg/g in the RG cultivar when mixed with roasted JP. The results indicate that the amount of bioaccessible phenolics ranged from 485 to 252 µg/g in potato cultivars mixed with roasted JP. The bioaccessibility of flavonoids ranged from 185.1 to 59.25 µg/g. The results indicate that the YG cultivar mixed with JP and SD showed the highest phenolic and carotenoid bioaccessibility. In contrast, the PM mixed with JP and SD contained the lowest phenolic and carotenoid bioaccessibility. Our results indicate that the highest flavonoid bioaccessibility occurred in R/R mixed with roasted JP and SD. The lowest flavonoids bioaccessibility occurred in PM and the RG. The maximum bioaccessible amount of capsaicin was observed in YG mixed with JP, while the minimum bioaccessibility was observed with PM.

## 1. Introduction

Crops such as potatoes (*Solanum tuberosum* L.) and peppers (*Capsicum annuum* L.) are among the most common food crops in human diets; they are members of the Solanaceae family [1,2]. These crops have been excellent sources of various phytochemical compounds, such as vitamins, phenolics, carotenoids, and capsaicinoid compounds [3,4,5]. Due to the abundance of bioactive compounds, these crops exhibit many nutritional and health benefits when consumed at appropriate levels [6,7], making these foods highly desirable choices for frequent consumption. The biological effects of bioactive compounds such as antioxidant activity, antimicrobial, and anticancer depend on the amount of those components consumed in the daily diet and their bioaccessibility [8]. These compounds vary widely in their chemical structures and biological functions; their bioaccessibility is not well understood [9]. The bioactive compound’s concentration and stability during food processing depend on several factors: the type and cultivar, growing conditions, geographical location, postharvest handling, processing conditions, and cooking methods [10,11,12]. It is important to obtain information about their bioaccessibility from the foods matrix and the factors that influence their bioaccessibility [13]. The bioaccessibility of bioactive compounds describes that part of the compound released from the food matrix and becomes available for absorption [14,15]. The combination of various food ingredients and cooking methods may significantly influence the bioaccessibility of bioactive compounds. Several studies have reported that it is possible to estimate the bioaccessibility of bioactive compounds by evaluating the quantity transferred to the micelle fraction following a simulated in vitro digestion procedure [15]. Several methodologies may be used to assess bioactive compounds’ bioaccessibility; among them, the most common is in vitro digestion [16]. The bioaccessibility of bioactive compounds depends on various factors, including their abundance within the food matrix, the heat of processing, and food additives such as dietary fat, oil, and certain enzymes [10,15]. Processing methods such as boiling, roasting, drying, and frying have been observed to enhance bioaccessibility significantly [15,17]. Cooking methods of food can influence bioactive compounds’ bioaccessibility, mainly through change and the disruption of the cell wall structure, leading to the release of these compounds, which implies higher bioaccessibility [17,18].

A few studies have reported the bioaccessibility of bioactive compounds in mixed diets, although only limited information is available. Potatoes and peppers are some of the most frequently consumed food sources globally, providing significantly greater amounts of bioaccessible bioactive compounds [19]. There is a significant variation of phytochemical compounds such as phenolics, flavonoids, anthocyanins, and carotenoids among potato cultivars [18,20]. These compounds exhibited significant antioxidant, anti-glycemic, antiviral, anticarcinogenic, and anti-inflammatory activities and showed anti-allergic and antimicrobial properties [21,22].

Potato and peppers are often consumed as part of an elaborate meal, including such ingredients such as salt, fiber, protein, and fat. Thus, it is possible that these ingredients influence the bioaccessibility of bioactive compounds. This study investigates the bioaccessibility of phenolics, flavonoids, carotenoids, and capsaicinoid compounds in cooked potato cultivars mixed with two roasted pepper varieties via in vitro digestion experiments.

## 2. Materials and Methods

### 2.1. Chemicals

Folin Ciocalteu Reagent (FCR), sodium carbonate, gallic acid, potassium chloride, sodium acetate, quercetin, lutein, and all digestive enzymes (α-amylase, pepsin, pancreatin, and porcine bile extract) were purchased from Sigma-Aldrich (Saint Louis, MO, USA). HPLC-grade acetonitrile butyl alcohol, methyl-t-ethyl ether (MtBE), and methanol reagents were obtained from Thermo Fisher Scientific Inc (Waltham, MA, USA).

### 2.2. Pepper and Potato Cultivars

Peppers in this study were sourced from Arkansas Valley Research Center, Rocky Ford CO (AVRC) in July 2016. Two different pepper varieties, Joe Parker (JP; hot) and Sweet Delilah (SD; sweet), were used in this study. Pepper fruits were transported to the San Luis Valley Research Center, Center CO (SLVRC), after harvest to complete the bioaccessibility experiments. Four different potato cultivars—Purple Majesty (PM; purple flesh), Yukon Gold (YG; yellow flesh), Rio Grande Russet (RG; white flesh), and a numbered line (CO 97226-2R/R (R/R; red flesh))—were used in this study. The potato cultivars were harvested at the end of the 2015 growing season at SLVRC.

### 2.3. Cooking Methods

Five to six pepper pods of each variety were washed, dried, and cut into small pieces with peduncles removed, then placed on an oven tray, transferred to a preheated oven set at 150 °C, and roasted for 20 min in a conventional oven. The oven had been preheated for uniform heat distribution. Once removed from the oven, all sample pieces were cooled, freeze-dried (LABCONCO, New York, NY, USA), ground, and stored at −20 °C until further analysis. Five potato tubers from each cultivar were collected at random, and the tubers were pierced twice on each side with a fork and baked at 204 °C for one hour in a commercial oven. All potato samples were then cut into small pieces, freeze-dried, ground using a coffee grinder, and stored at −20 °C until further analysis.

### 2.4. Extraction and Estimation of the Total Phenolics and Flavonoids

Total phenolics and total flavonoids were extracted with pure methanol. To 15 mL of 100% methanol, 0.4375 g of potato freeze-dried powder and 0.0625 g of roasted pepper freeze-dried powder were added and homogenized for 5 min. Supernatants of pepper and potato extract were filtered to evaluate total phenolics and total flavonoids. The total phenolic content was calculated according to the method declared by [20] with modifications. FCR solution was added to pepper and potato extract, and sodium carbonate was added to the 96 microplates. The total phenolic content of pepper and potato samples was calculated as gallic acid equivalents (μg/g). A colorimetric method was used to evaluate the total flavonoid content in pepper samples. Aluminum chloride was added to the pepper extract in 96 microplates. The total flavonoid content of pepper samples was expressed as quercetin equivalents (μg/g).

### 2.5. Extraction and Estimation of Total Carotenoids

Total sample carotenoids were extracted with water-saturated butanol. To 15 mL of water-saturated butanol, 0.4375 g of potato freeze-dried powder or 0.0625 g of freeze-dried powder of roasted peppers was added and homogenized for 5 min. The mixture was covered with aluminum foil and allowed to stand in the fume hood for 60 min at room temperature. The absorbance of the mixture was measured at 450 nm. Lutein was used as the standard, and total pepper carotenoids were quantified as μg of lutein equivalent per gram of dry weight materials using a 5-point calibration curve with an R^2^ value of 0.996.
(1)$$\begin{array}{c} \mathrm{Total}\;\mathrm{Carotenoids}\\ \left(\mathrm{mg}/g\right) \end{array}\;=\;\frac{\left(\frac{\mathrm{Sample}\;\mathrm{Abs}\;-\;y\;\mathrm{intercept}}{\mathrm{slope}}\right)\;\times \;\mathrm{Orig}\;\mathrm{Vol}\;(L)\;\times \;\begin{array}{c} \mathrm{Dilution}\\ \mathrm{Factor} \end{array}\;}{\mathrm{Sample}\;\mathrm{Weight}\;(g)}$$
where: Sample Abs = Sample Absorbance; Orig Vol = Original Volume for sample preparation, in liters; Slope = Slope from standard curve.

### 2.6. Extraction of Capsaicinoid Compounds

Extraction and quantification of capsaicinoid compounds were performed as described by [12]. A total of 0.500 mg of freeze-dried ground pepper was added to a 15 mL polypropylene tube. Ten milliliters of methanol was added to each sample and placed in an orbital shaker overnight at 25 °C. The supernatant was transferred to a fresh 15 mL tube. The sample was re-extracted with 10 mL more of methanol, and the resulting supernatants were combined. One milliliter of the methanolic extract was filtered through a 0.45 µm filter cartridge (Advanced Microdevices, Ambala, India) and poured into a 1.8 mL sample glass vial for HPLC analysis.

### 2.7. Analysis of Capsaicinoid Compounds

Capsaicin and dihydrocapsaicin were quantified using a Waters (Milford, MA, USA) HPLC system equipped with a fluorescence detector and a Waters Nova-Pak C18 4 µm, 4.6 × 150 mm C18 column. Aqueous methanol A (10% methanol) and B (100% methanol) were used as eluent with a flow rate of 0.4 mL/min, and a gradient of 0 to 10 min, 80% A and 20% B. The fluorescence detector was set to an excitation wavelength of 280 nm and an emission wavelength of 338 nm. Levels of capsaicin and dihydrocapsaicin were estimated using a calibration curve with a standard of capsaicin and dihydrocapsaicin with concentrations ranging from 0.1 to 10 μg/mL. R^2^ values were obtained from the standard curve of capsaicin and dihydrocapsaicin with 0.995 and 0.997, respectively.

### 2.8. In Vitro Digestion

The in vitro digestion protocol described by [23] was performed with modifications in triplicate. In a 50 mL polypropylene tube, 0.4375 g of potato cultivar or 0.0625 g of roasting peppers were mixed with 5 mL of distilled water and homogenized by vortex for 30 s. For salivary digestion, samples were treated with 1 mL α-amylase solution with enzymatic activity 24.0–36.0 U/mg (70 mg/mL in phosphate buffer of 0.1 M pH = 6.9) at 37 °C for 10 min. Sample pH was adjusted to pH 2 with 0.1 M HCl. Samples were then treated with 0.3 mL pepsin with enzymatic activity ≥ 3200 U/mg (300 mg/mL HCl-KCl, 0.2 M pH 1.5) in a 37 °C water bath to complete the gastric digestion phase. Samples were treated with pancreatin with enzymatic activity as markers (5 mg/mL, phosphate buffer, 0.1 M, pH 7.5) and 3.3 mL porcine bile extract (17.5 mg/mL phosphate buffer, 0.1 M, pH 7.5) in a 37 °C water bath to complete the intestinal digestion phase. The resulting digestates were centrifuged at 5000× *g* for 20 min. Supernatants were collect and stored at −80 °C until further analysis. The bioaccessibility of bioactive compounds in this study is expressed as the percentage of bioactive compounds transferred to the aqueous phase during the in vitro digestion process (amounts of total compounds in the aqueous phase/amount compounds in the sample × 100).

## 3. Statistical Analysis

All experiments were carried out in triplicate, and the data were subjected to analysis of variance (ANOVA); Tukey’s test was performed to determine significance at *p* < 0.05 between treatments. All statistical analyses were performed with R software version 3.4.3 for Windows.

## 4. Results and Discussion

### 4.1. Total Phenolics and Bioaccessibility

Vegetables and fruits are the primary sources of dietary polyphenolics, such as phenolic acids, hydroxycinnamic acid derivatives, and flavonoids. These compounds provide many nutritional and health benefits, including antioxidant, anti-inflammatory, and antimicrobial activities [5,20,24]. Thus, phenolic compounds are recognized as rich sources of dietary antioxidants [3]. However, their health benefits correlate with their bioaccessibility [25]. Phenolic compounds such as chlorogenic acid, neochlorogenic acid, cryptochlorogenic acid, caffeic acid, p-coumaric acid, and ferulic acid are present at high concentrations in colored flesh potato cultivars and quercetin, luteolin, and capsaicinoids in pepper varieties; however, the bioaccessibility of these compounds can be highly variable.

The levels of phenolics and bioaccessible phenolics in potato cultivars mixed with roasted peppers are shown in Figure 1. Red-fleshed potato selection R/R mixed with roasted JP had the maximum amount of phenolics (2164 µg/g) followed by PM (1621 µg/g), YG (644 µg/g), and RG (339 µg/g). Similar levels of phenolics (2316 to 329 µg GAE/g FD) in the same order were obtained when mixed with roasted SD (Table 1). Higher levels of total phenolics were found in red and purple-fleshed tubers than in white and yellow cultivars [20]. These differences in total phenolic content could be attributed to various factors, including the variety, flesh color, starch content, and maturity stages [26].

After in vitro digestion, a significant reduction of total phenolics was observed in all samples, which agrees with an earlier report by Andre et al. (2015) [27]. The bioaccessible phenolic content was 1316, 869, 485, and 252 µg/g in R/R, PM, YG, and RG potatoes, respectively, when mixed with JP (Figure 1a, and Table 1). Our results indicate that 53–75% of phenolic compound content was released from potato cultivars mixed with JP; by contrast, the release was 53–88% in potatoes cultivars mixed with SD (Figure 1b and Table 1). Our results show that the highest release of the phenolic compounds was observed in YG mixed with SD, whereas the lowest release of phenolic compounds was observed in PM mixed with JP. During gastric digestion, phenolic compounds were highly bioaccessible; 43.8–93.73% of phenolic compounds were released [7]. Several reports have established that the bioaccessibility of bioactive compounds is influenced by the composition of the digested food matrix and physicochemical properties, such as pH, temperature, and texture of the matrix [13,27]. 

### 4.2. Total Flavonoids and Bioaccessibility

Flavonoids are the most common group of polyphenolic compounds in plants. Flavonoids are natural polyhydroxylated compounds with a proven positive impact on human health. The impact of dietary flavonoids depends on their bioaccessibility. There is little published information on the bioaccessibility of flavonoid compounds following the in vitro digestion procedure.

The results in Figure 2 show the levels of flavonoids and bioaccessible flavonoids in different potato cultivars either mixed with roasted JP pepper (Figure 2a) or with roasted SD (Figure 2b). A significant variation (*p* ≤ 0.05) was observed in the content of total flavanoids between different potato cultivars mixed with roasted JP (222.4 to 59.25 µg QE/g FD) and SD (277 to 83.9 µg QE/g FD) before digestion (Table 1). Similarly, the bioaccessibility of flavonoid compounds ranged from 185 to 59 µg QE/g FD and from 231 to 64 µg QE/g FD for potato cultivars mixed with JP and SD, respectively. The total flavonoid content and bioaccessibility of flavonoid compounds among the potato cultivars we tested was in the order R/R > PM > RG > YG, irrespective of peppers in the study (Table 1 and Figure 2). Our results show that the release of flavonoid compounds ranged from 65 to 83% in potato cultivars mixed with JP, whereas this release ranged from 58 to 83% in potato cultivars mixed with SD. Previous studies have shown that purple and red cultivars had twice the flavonoid concentration of white cultivars [28]. Various studies suggested that variations in flavonoid levels are primarily the result of the diversity of genotypes, landraces, varieties, and the ripening stage of the fruits [29]. Several studies reported that flavonoid bioaccessibility was dependent on the digestible and non-digestible fibers in the tested food product [30]. Previous studies reported that thermal treatments during food processing increased bioactive compounds’ bioaccessibility [31]. 

### 4.3. Total Carotenoids and Bioaccessibility

Carotenoids are a large class of bioactive compounds responsible for the attractive color of many fruits and vegetables. Due to the pro-vitamin A activity, carotenoids constitute a significant source of antioxidants associated with health benefits [32]. Carotenoid bioaccessibility depends on the degree of food processing and matrix composition [33].

Yellow flesh cultivars have generally shown a much higher average of total carotenoid content when compared to red and purple-fleshed potatoes [34]. Our studies show that carotenoid levels (194 to 43 µg Lu/g FD) and carotenoid bioaccessibility (152 to 30 µg Lu/g FD) varied in four potato cultivars when mixed with JP (3a). Higher levels of carotenoids were present in YG and R/R mixed with JP compared to PM and RG. Similarly, total carotenoid levels are between 230 and 49 µg Lu/g FD, and the bioaccessibility of carotenoids ranged from 185 to 17 µg Lu/g FD in potato cultivars when mixed with roasted SD (3b). There are no significant differences in the % release of carotenoids between JP and SD except the cultivar effect (Figure 3 and Table 1). The differences in the release of flavonoids and carotenoids between JP and SD could be explained by several factors, such as the physical and chemical nature of the food matrix, cooking methods, solubility, and polarity of carotenoids, and interaction with other compounds during the digestion procedure.

Several studies reported that the differences in total carotenoid content among samples have been attributed to variety, maturity stage, and cooking methods, and the presence of other nutrients such as fat and fiber [18,35,36].

Andre et al. [27] reported that the bioaccessibility of lutein and zeaxanthin in the yellow clones ranged from 76 to 82% for lutein and from 24 to 55% for zeaxanthin. The bioaccessibility of carotenoids from raw, frozen, and boiled red chili peppers was studied by Pugliese et al. [11]. They reported that b-carotene and b-cryptoxanthin had lower bioaccessibility, while capsanthin, zeaxanthin, and antheraxanthin had higher bioaccessibility. O’Sullivan et al. [37] have demonstrated that carotenoid bioaccessibility from red bell peppers ranged from 33 to 87%. One report indicated that the percent accessible all-trans-β-carotene in the supernatant phase was significantly higher—between 24 and 41%—without fat and between 28 and 46% with fat [38]. Several studies have indicated that carotenoid bioaccessibility strongly depends on the food matrix characteristics, chemical structure of carotenoids, and thermal treatments during food processing [32]. Various studies suggested that cooking methods such as roasting increase the accessibility of carotenoids [39]. The interaction bar of the effects of potato cultivars and roasted peppers on carotenoid compounds is shown in Figure 6c. The bar graph suggests a statistically significant interaction.

### 4.4. Capsaicinoid Compounds and Bioaccessibility

Capsaicinoid compounds are widely distributed in pungent pepper fruits. They are the primary active component in chili peppers and have known health benefits. Capsaicin and dihydrocapsaicin are the most abundant capsaicinoids in peppers, together constituting about 90% of the total capsaicinoids in peppers [12]. In recent years, the consumption of pungent components in hot peppers has increased due to their associated benefits to human health [40]. Figure 4 shows a chromatogram of capsaicinoids in JP.

Little information is available in the literature on the bioaccessibility of capsaicinoid compounds in mixed diets. It is vital to assess any beneficial effect when capsaicinoid compounds are added to a carbohydrate-rich diet. The levels of capsaicin and bioaccessible capsaicin in potato cultivars mixed with hot roasted pepper are shown in Figure 5. The range of capsaicin and bioaccessible capsaicin in potato cultivars mixed with roasted JP was 52.9 to 66.8 µg/g FD, whereas the levels of dihydrocapsaicin and bioaccessible dihydrocapsaicin in potato mixed with roasted JP ranged from 22 to 13 µg/g FD, whereas potato cultivars mixed with roasted SD did not show capsaicin and dihydrocapsaicin. After in vitro digestion, there was a significant reduction in capsaicin and dihydrocapsaicin in potato cultivars mixed with roasted JP. Similarly, loss of capsaicinoid compounds was observed in jalapeño peppers following heat treatment [17]. The bioaccessibility of capsaicin ranged from 55 to 72%, while dihydrocapsaicin’s bioaccessibility ranged from 56 to 83%. Potato cultivar YG mixed with hot roasted pepper had the highest capsaicin bioaccessibility, while cultivar PM had the lowest. The dihydrocapsaicin bioaccessibility ranking was: RG mixed with hot pepper > YG > R/R > PM. Our in vitro digestion results support those reported by [17]. They found that cooking methods such as boiling and grilling improved the bioaccessibility of dihydrocapsaicin and capsaicin, respectively, in red pepper. For instance, capsaicin bioaccessibility is significantly influenced by the interaction between ripening stage and heat processing, whereas dihydrocapsaicin bioaccessibility is substantially influenced by the interaction between the type of dietary fat, ripening stage, and heat processing [17].

The interaction of the effects of potato cultivars and roasted peppers on phenolics is shown in Figure 6a, flavonoids 6b and carotenoids 6c. The interaction bar graph (with a difference as a response) showed that there is evidence of a significant interaction between potato cultivars and roasted pepper bioactive compounds. 

## 5. Conclusions

Peppers and potatoes are the most consumed crops in the world due to culture and eating habits. Significant variations (*p* ≤ 0.05) were observed in the levels of bioactive compounds in potatoes mixed with roasted peppers. After in vitro digestion, our results indicate that more than 50% of bioactive compounds are released from the matrix. The present study suggests that phenolics, flavonoids, carotenoids, and capsaicinoids are highly bioaccessible from potato cultivars mixed with roasted pepper varieties. Phenolic compound levels are high in the R/R cultivar, but the highest release of the phenolic compounds was observed in YG mixed with SD, whereas the lowest release of phenolic compounds was observed in PM mixed with JP. Similarly, red flesh cultivars have higher flavonoid levels, and there were differences between JP and SD in % release after in vitro digestion. There were no significant differences in the % release of carotenoids between JP and SD except the cultivar effect.

## Figures and Tables

**Figure 1 foods-10-01849-f001:**
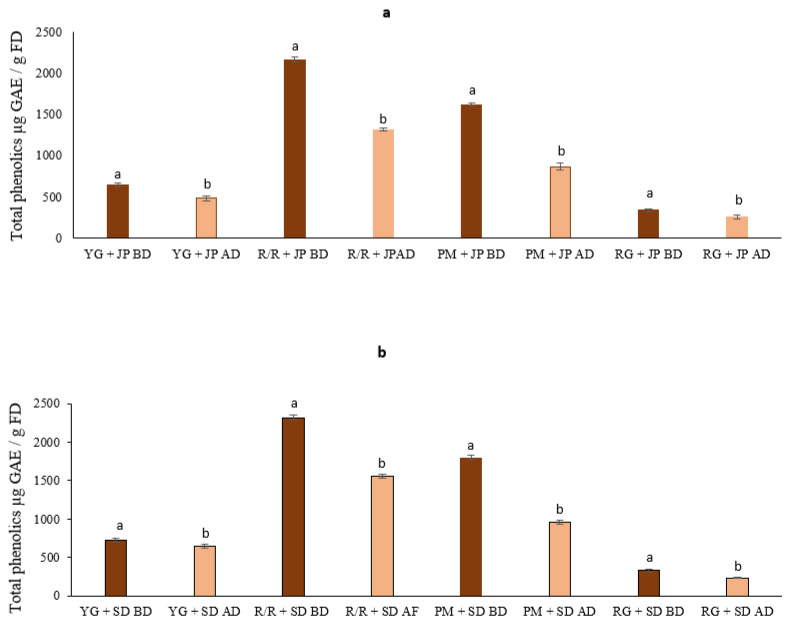
The levels of total phenolics and bioaccessibility of cooked potato varieties mixed with roasted JP (**a**) and with roasted SD (**b**). Data are the mean of three replicates with standard deviation and are expressed as per gram of freeze-dried weight. Significant differences are denoted by different letters, while the same or shared letters indicate that they are not significant to each other. BD and AD: before and after digestion.

**Figure 2 foods-10-01849-f002:**
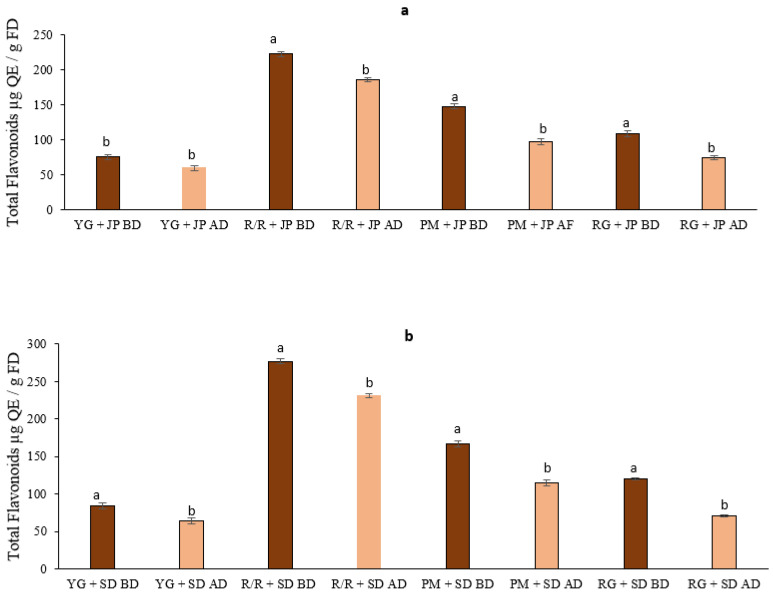
The levels of total flavonoids and bioaccessibility of cooked potato cultivars mixed with roasted JP (**a**) and roasted SD (**b**). Data are the mean of three replicates with standard deviation and expressed as per gram of freeze-dried weight. Significant differences are denoted by different letters, while the same or shared letters indicate that they are not significant to each other. AD and BD: before and after digestion.

**Figure 3 foods-10-01849-f003:**
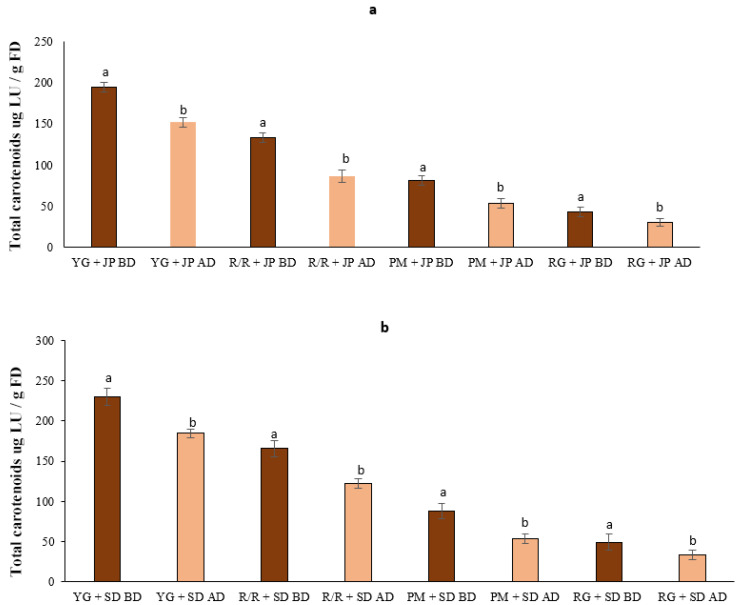
The levels of total carotenoids and bioaccessibility of cooked potato cultivars mixed with roasted JP (**a**) and roasted SD (**b**). Data are the mean of three replicates with standard deviation and are expressed as per gram of freeze-dried weight. Significant differences are denoted by different letters, while the same or shared letters indicate that they are not significant to each other. BD and AD: before and after digestion.

**Figure 4 foods-10-01849-f004:**
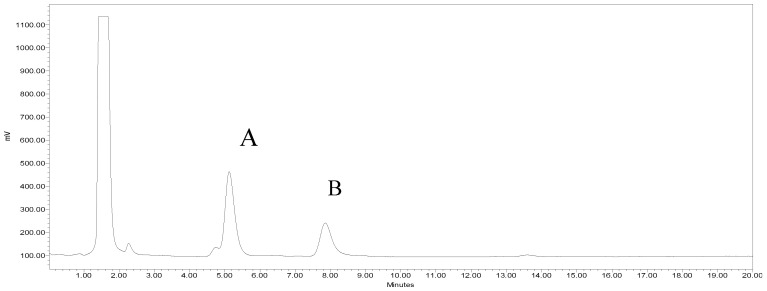
A representative HPLC chromatogram of Joe Parker showing baseline separation of capsaicin (**A**) and dihydrocapsaicin (**B**).

**Figure 5 foods-10-01849-f005:**
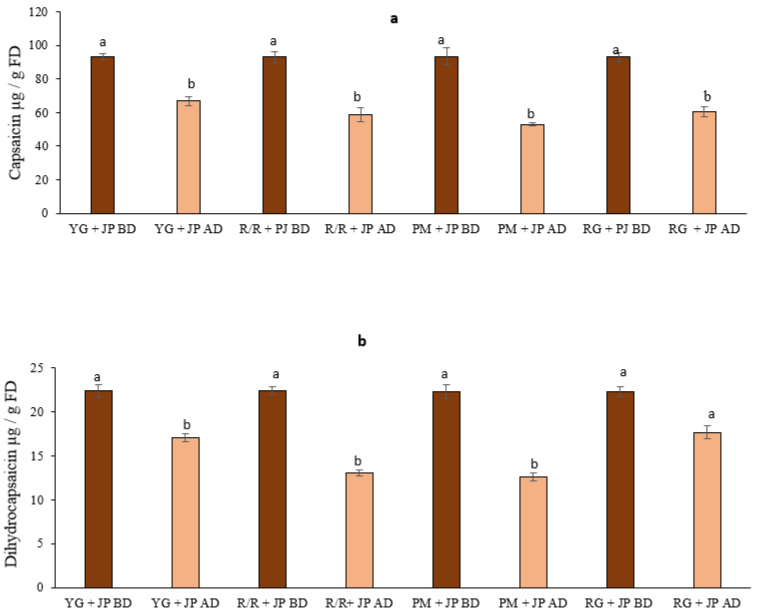
The levels of capsaicin and bioaccessibility of capsaicin of cooked potato cultivars mixed with roasted JP (**a**) and the levels of dihydrocapsaicin and bioaccessibility of dihydrocapsaicin of cooked potato cultivars mixed with roasted JP (**b**). Data are the mean of three replicates with standard deviation and are expressed as per gram off freeze-dried weight. Significant differences are denoted by different letters, while the same or shared letters indicate that they are not significant to each other. BD and AD: before and after digestion.

**Figure 6 foods-10-01849-f006:**
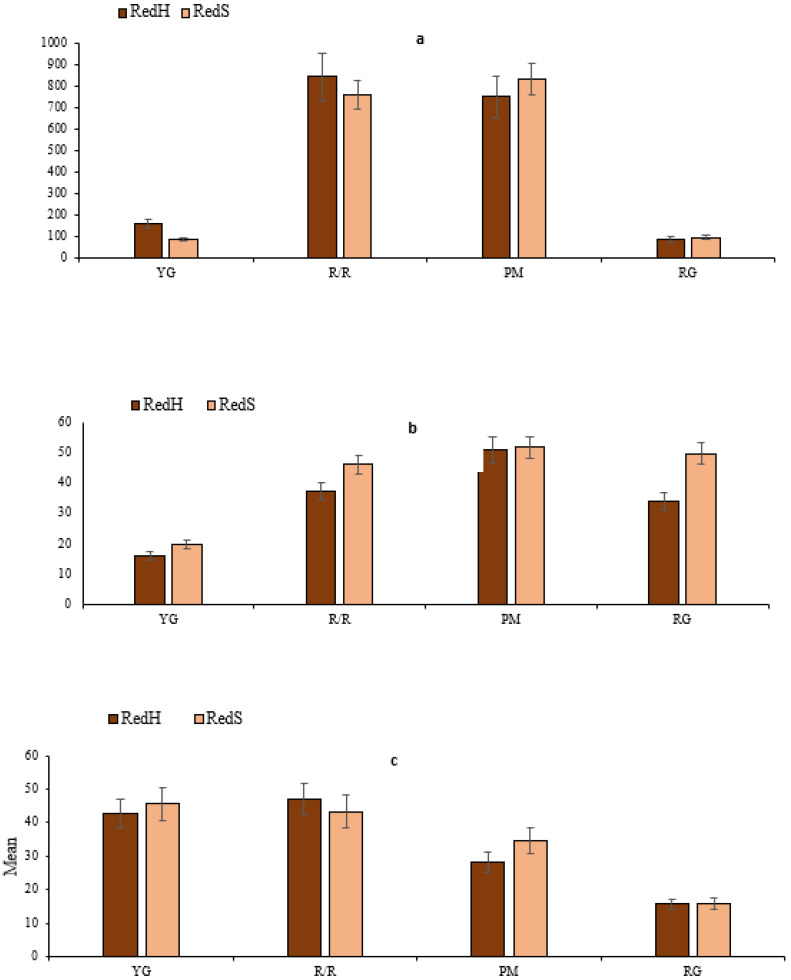
(**a**–**c**) The interaction bar of cooked potato cultivars with roasted pepper cultivars on bioactive compounds.

**Table 1 foods-10-01849-t001:** Concentration of total bioactive compounds in potatoes cultivars mixed with two roasted peppers, bioaccessibility, and percentage release during in vitro digestion experiment. Data are the mean of three replicates with standard deviation and are expressed as per gram of freeze-dried weight. Significant differences between bioactive compounds before (^a^) and after digestion (^b^) are denoted by different letters, while the same or shared letters indicate that they are not significant.

Treatment 1	Bioactive Compounds								
	Total Phenolic µg GAE/g FD	Total Bioaccessible Phenolics µg GAE/g FD	% Release	Total Flavonoids µg QE/g FD	Total Bioaccessible Flavonoids µg QE/g FD	% Release	Total Carotenoids µg Lu/g FD	Total Bioaccessible Carotenoids µg Lu/g FD	% Release
YG + JP	644.3 ^a^ ± 27	485.7 ^b^ ± 29	75.3	75.3 ^a^ ± 3.6	59.25 ^b^ ± 4.1	78.6	194.3 ^a^ ± 5.6	151.7 ^b^ ± 5.6	78
R/R + JP	2164.3 ^a^ ± 29	1316.8 ^b^ ± 20	60.8	222.4 ^a^ ± 3.5	185.18 ^b^ ± 2.9	83.2	133.2 ^a^ ± 5.5	86.1 ^b^ ± 11.3	64.6
PM + JP	1620.8 ^a^ ± 19	869.06 ^b^ ± 42	53.6	147.6 ^a^ ± 3.7	96.7 ^b^ ± 3.8	65.5	81.6 ^a^ ± 5.7	53.4 ^b^ ± 2.5	65.4
RG + JP	338.7 ^a^ ± 20	252.7 ^b^ ± 22	74.6	107.9 ^a^ ± 4.3	74.07 ^b^ ± 2.9	68.6	42.9 ^a^ ± 5.4	30.4 ^b^ ± 1.8	71
	**Bioactive Compounds**								
**Treatment 2**	**Total Phenolic µg GAE/g FD**	**Total Bioaccessible Phenolics µg GAE/g FD**	**% Release**	**Total Flavonoids µg QE/g FD**	**Total Bioaccessible Flavonoids µg QE/g FD**	**% Release**	**Total Carotenoids µg Lu/g FD**	**Total Bioaccessible Carotenoids µg Lu/g FD**	**% Release**
YG + SD	723.6 ^a^ ± 19	639.4 ^b^ ± 26	88.3	83.9 ^a^ ± 3.6	64.1 ^b^ ± 3.5	76.4	230 ^a^ ± 11.1	184.5 ^b^ ± 5.3	80.1
R/R + SD	2316.3 ^a^ ± 32	1556.3 ^b^ ± 26	67.1	277 ^a^ ± 3.7	231.1 ^b^ ± 2.9	83.4	165.5 ^a^ ± 9.6	122.2 ^b^ ± 5.6	73.8
PM + SD	1794.3 ^a^ ± 29	960 ^b^ ± 28	53.4	167.1 ^a^ ± 5.2	114.8 ^b^ ± 3.3	68.9	88.09 ^a^ ± 9.1	53.4 ^b^ ± 5.7	60.6
RG + SD	328.7 ^a^ ± 20	234.6 ^b^ ± 11	71.3	120.2 ^a^ ± 2.5	70.6 ^b^ ± 1.5	58.7	49.3 ^a^ ± 9.3	17.3 ^b^ ± 2.5	68.3

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
