# Peer review of "The Bioaccessibility of Phenolics, Flavonoids, Carotenoids, and Capsaicinoid Compounds: A Comparative Study of Cooked Potato Cultivars Mixed with Roasted Pepper Varieties"

_foods, 2021, doi:10.3390/foods10081849_

Round 1
Reviewer 1 Report
This is opinion of the reviewer that this paper cannot be accepted by Foods. My individual comments are listed below.
Language is not acceptable. This paper must be corrected by a native speaker in English who is a chemist.
The renowned journal should accept only papers with an HPLC analysis of individual phenolic compounds.
The citations must be in the style of Foods.
The references must be prepared according to the Foods style.
Table 1 – The significance of differences must be completer. The results must be reported with ±SD.
Figure 5 – The results must be reported as bars with error bars.
16 – “Total phenolics, flavonoids, carotenoids” are the terms of analytical chemistry. The term of “bioaccessibility of total phenolics” is wrong.
Title, l. 40 – Flavonoids belong to phenolics.
89 – The enzyme activity must be reported.
155 – “linear coefficient constants”?
180 – Phenolic acids are a single phenolics not polyphenolics.
Typical chromatogram of capsaicinoids should be reported.
Author Response
Reviewer 1
This is opinion of the reviewer that this paper cannot be accepted by Foods. My individual comments are listed below. Language is not acceptable. This paper must be corrected by a native speaker in English who is a chemist.
This manuscript was edited by fellow faculty who is a native speaker in English also a biochemist.
The renowned journal should accept only papers with an HPLC analysis of individual phenolic compounds.
In our study, we have used the HPLC instrument with fluorescence detector to evaluate the Capsaicinoid compounds (capsaicin and dihydrocapsaicin) details were given in section 2.7 in the materials and method section.
Specific comments:
The citations must be in the style of Foods
We changed all the citations in our manuscript to foods journal style.
The references must be prepared according to the Foods style.
We changed all the references in our manuscript to foods journal style.
Table 1 – The significance of differences must be completer. The results must be reported with ±SD.
We added the significance of the differences to Table 1 with ±SD.
Figure 5 – The results must be reported as bars with error bars.
The figures (5a,5b, and 5c) have been modified as suggested.
Line 16 – "Total phenolics, flavonoids, carotenoids" are the terms of analytical chemistry. The term of "bioaccessibility of total phenolics" is wrong
The statement on bioaccessibility of total phenolics modified to phenolics, flavonoids and carotenoids (line 16).
Title, l. 40 – Flavonoids belong to phenolics.
We revised the statement (line 40).
Line 89 – The enzyme activity must be reported.
We added the enzyme activity for the enzymes used in the study in the methods section 2.8 (lines 155-156; 158; and 160-161).
Line 155 – "linear coefficient constants"?
We revised the statement in the method section (Lines 149-150).
Line 180 – Phenolic acids are a single phenolics not polyphenolics.
We revised the statement as suggested (lines 179-181).
Typical chromatogram of capsaicinoids should be reported.
We explained the chromatogram of capsaicinoids in methods section 2.7 (lines 141-150), and representative high-performance liquid chromatography (HPLC) chromatogram of Joe Parker was added to the manuscript (figure 4) (lines 292-294).
Reviewer 2 Report
The authors report on the bioaccessibility of phenolics, flavonoids, c arotenoids, and capsaicinoid compounds in different cooked potatoes mixed with roasted peppers. The topic is interesting and overall the research is well conducted.
Certain points need to be addressed:
I do not believe the in vitro digestion model is suitable to assess the digestibility of complex food matrices such as the ones analysed in the present study. Similarly, it is difficult to assess the impact of processing on bioaccessibility using this in vitro digestion model.
The authors report no significant differences in the % release of carotenoids between JP and SD. How do the authors explain this trend, rather different from the flavonoids?
What is the activity of the enzymes used (salivary amylase, pepsin and pancreatin)?
Author Response
Reviewer: 2
The authors report on the bioaccessibility of phenolics, flavonoids, carotenoids, and capsaicinoid compounds in different cooked potatoes mixed with roasted peppers. The topic is interesting and overall, the research is well conducted.
Certain points need to be addressed:
I do not believe the in vitro digestion model is suitable to assess the digestibility of complex food matrices such as the ones analyzed in the present study. Similarly, it is difficult to assess the impact of processing on bioaccessibility using this in vitro digestion model.
We have used the in-vitro digestion model, which is frequently reported in the scientific literature, to evaluate bioactive compounds' bioaccessibility.
Recent references are listed here
1. Khoja, K.; Buckley, A.; F. Aslam, M.; A. Sharp, P.; Latunde-Dada, G.O. In Vitro Bioaccessibility and Bioavailability of Iron from Mature and Microgreen Fenugreek, Rocket and Broccoli. Nutrients 2020, 12, 1057. https://doi.org/10.3390/nu12041057. (Simulated gastrointestinal in vitro digestion and subsequent measurement of ferritin in Caco-2 cells)
2. López-López, A.; Moreno-Baquero, J.M.; Garrido-Fernández, A. In Vitro Bioaccessibility of Ripe Table Olive Mineral Nutrients. Foods 2020, 9, 275. https://doi.org/10.3390/foods9030275
3. Andre, C.M., Evers, D , Ziebel, J., Guignard, C., Hausman, J., Bonierbale, M., Felde, T Z., Burgos, G. In Vitro Bioaccessibility and Bioavailability of Iron from Potatoes with Varying Vitamin C, Carotenoid, and Phenolic Concentrations. J. Agric. Food Chem, 2015, 63, 9012−9021.
4. Miranda, L.; Deuber, H.; Evers, D. The impact of in vitro digestion on bioaccessibility of polyphenols from potatoes and sweet potatoes and their influence on iron absorption by human intestinal cells. Food Funct, 2013 4,1595-1601.
The authors report no significant differences in the % release of carotenoids between JP and SD. How do the authors explain this trend, rather different from the flavonoids?
We revised and modified the statement (lines 267-270).
What is the activity of the enzymes used (salivary amylase, pepsin and pancreatin)?
We added the enzyme activity for the enzymes used in the study in the methods section 2.8 (lines 155-156; 158; and 160-161).
Round 2
Reviewer 1 Report
The authors corrected this paper properly taken under considerations all my comments. Therefore, I can accept it now.
Reviewer 2 Report
The authors have addressed the comments.